# Knowledge and coping style about depression in medical students: A cross-sectional study in China

**Yajun Lian**[1☯]**, Yumeng Yan**[2☯]**, Weiwei Ping**[3]*****, Zhiyong Dou**[4]**, Xiaoyan Wang**[5]**, Hui Yang**[3]

**1** Department of General Medicine, Heping Hospital Affiliated to Changzhi Medical College, Changzhi, China, **2** School of Public Health, Shanxi Medical University, Taiyuan, China, **3** Department of Public Health and Preventive Medicine, Changzhi Medical College, Changzhi, China, **4** Department of Physical Examination Center, Heping Hospital Affiliated to Changzhi Medical College, Changzhi, China, **5** Department of Ultrasound B, Heping Hospital Affiliated to Changzhi Medical College, Changzhi, China

☯ These authors contributed equally to this work.
* weiweip@czmc.edu.cn

**Data Availability Statement:** All relevant data are within the paper and its Supporting information files.

**Funding:** the study funded by SHANXI office for education science planning (GH-21565). The

## Abstract

### Objectives

The current study aimed at ascertaining the depression levels of medical students and their knowledge levels of depression, and exploring the relationship between the level of knowledge and coping styles of the medical students on depression.

### Methods

An online-based survey was developed in Changzhi Medical College. The questionnaire included demographic and socioeconomic data, questions about depression knowledge and copying styles of depression, and the Zung Self-Rating Depression Scale (SDS). A total of 1931 questionnaires were returned by respondents.

### Results

The medical students produced a mean SDS score of 44.29 (SD = 11.67). The prevalence of depression was 29.7%. Sophomore, female, and poor family relationships were parameters associated with a higher SDS score. The total correct rate for knowledge of depression was 64.14%. There were statistical differences between with depression students and non-depression students on the rate of the correct answers in the following questions:"Female has more probability", "Depression can be adjusted by oneself", "Associated with one's character", "Know cure method of depression", "Know drug use of depression", "Know depression influence for health", and "Know prevention method of depression". Depression students were more likely to have a lower rate of correct answer for above questions. Asking for help from psychological consultation was the primary coping mechanism among the medical students. The logistic regression analysis results found that depressed students who chose the coping way of no ways of coping were more likely to be females OR = 1.470 (1.078, 2.005), residents in rural area OR = 1.496 (1.038, 2.156), in poor family relationships

funders had no role in study design, data collection and analysis, decision to publish, or preparation of the manuscript.

**Competing interests:** The authors have declared that no competing interests exist.

OR = 2.428 (1.790, 3.293), and have lower cognitive level of depression knowledge OR = 1.920 (1.426, 3.226).

## Conclusions

It is necessary to focus on mental health of medical students, especially in female, residents in rural area, in poor family relationships, and having lower cognitive level of depression knowledge. Medical students were insufficient on depression knowledge and coping styles, and efforts that train students know risk of impaired mental health could also improve diagnosis and treatment.

## Introduction

Depression is an important public health problem with a significant effect on individual health. More than 264 million people suffered from depression in 2017, which was the leading cause of the global burden of diseases [1, 2]. Depression among the urban poor reached 12.30% in Peninsular Malaysia in 2013 [3], while a 17.96% prevalence of depressive symptoms was reported in the cohort study in China in 2019 [4].

There are rising concerns about depression of college students. The prevalence of college students with depression had increased rapidly from 10% to 15% from 2000 to 2006 [5]. The results during the COVID-19 pandemic showed 48.14% of American college students had moderate or severe level of depression [6]. Depressive disorder reached 18.5% in 14,371 first-year students from 19 colleges in 2018 [7].

Some studies revealed that compared with age-matched non-medical college students, medical students suffered a higher level of depression, as the learning environment and the training process contributed to depression in medical student [8]. In the systematic review from Europe and North America in 2014, 6.0%-66.5% of depression was reported in medical students. Progresses and changes through the course, female, low family income, debt burden, and regulatory responsibility for medical students could be risk factors for medical students [9]. Another systematic review from 47 countries reviewed by Rotenstein et al. showed that 27.2% of medical students experienced depressive disorder [10]. When suffering from depression, medical students were frequently reluctant to seek help. The study in US medical college in 2010 by Tjia found that only 12.5% of depressed students sought treatment, in whom 18.4% were diagnosed. Those who were 25 years or older, had a family history of depression or a previous history of depression had a higher rate of treatment. The barriers for under-treatment of depression may include the stigma associated with seeking mental health services, fear of a negative impact on one's job, and fear that a diagnosis would appear on ones academic records [11, 12]. The survey by Schwenk revealed that 70%-80% of medical students with moderate to severe depression had no history of treatment [13]. The study results from China in 2013 showed the rate of seeking medical care among depressed people was 17.3%, and the hospitalization treatment rate was only 2.40% [14]. Medical students' attitude toward using psychiatric services was related to correct knowledge about the etiology of depression [15].

Therefore, we found a conflict between the high rate of depression and the low rate of treatment. The key point is to distinguish depression in the population. To date, limited attention has been directed at individuals' knowledge, beliefs, and ways of coping with depression. Exploration of such knowledge and coping style(s) for depression may reveal factors related to the conflict.

The current study aimed at ascertaining the level of medical students depression and their levels of knowledge about depression, and then subsequently exploring the relationship between level of knowledge and coping style of the medical students on depression.

## Methods

### Participants

The questionnaire survey was conducted online in March 2 to March 20, 2021. Social media (e.g., We-chat) was the tool. Respondents have been recruited among students from Changzhi Medical College. The questionnaire included a statement about informed consent at the beginning. The participant was asked to answer the question "are you still willing to participate in our survey after reading and understanding our research intentions?" Only those who chose "yes" were considered consent to write out the questionnaire. During the period of work, 2200 medical students were invited. Each invited medical students could complete the questionnaire only once. By March 20, a total of 1931 questionnaires had been returned by respondents, and the response rate was 87.8%. 269 invited no returned the questionnaire, the reasons may be disagree or miss the time. Questionnaires that had completed less than 100 seconds were considered unusable, 1909 questionnaires were usable, and 22 were removed from the analysis.

### Survey questionnaire

The questionnaire included three parts.

The first part surveyed the demographic characteristics and selected characteristics of the respondents. The demographic characteristic variables were age and sex, while the socioeconomic variables included specialty, grade, region, place of residence, one-child family or not, and the family relationship. These variables were displayed in Table 1.

The second part of the questionnaire examined the knowledge of depression and copying styles of the respondents. Based on a literature review of depression, the survey team designed 18 questions. These questions included prevention, pathogenic factor, clinical symptoms, treatments of depression, and the coping strategy [10, 11, 14, 15]. The copying strategy was assessed by the question "if you were affected by depression, which copying style would be chosen (multiple choice)? " "No ways of coping", "ask friends for help", "search a book", "ask for help from psychological consultation in the college", and "see the doctor in a hospital" had been listed choices. If a respondent answered a question correctly, then a score of 1 was recorded, and the question was considered "known". If a respondent answered a question incorrectly, then a score of 0 was recorded, and the question was considered "unknown". A high knowledge level of depression was defined if the total score of respondents was more than 12. Otherwise, it was defined a low knowledge level of depression. The Cronbach's α was 0.745.

The final part of the survey comprised the Zung Self-Rating Depression Scale (SDS), created by William W.K [16]. The SDS is a reliable and valid tool to screen for depression and is often used to assess the psychological and somatic symptoms of depression [17]. It has been widely used in various age groups in the Chinese [18–21]. The SDS includes 20 items and is scored on a Likert scale evaluated by 1–4 score (a little of the time, some of the time, a large part of the time, most of the time). The raw sum score of the scale ranges from 20 to 80, and then, the raw score is converted into a 100-point scale as the Self Rating Depression Index. Higher index scores on the SDS reflects higher levels of depression symptoms: less than 50 score is no depression, 50–59 score is minimal to mild depression, 60–69 score is moderate to marked depression, and greater than 70 score is severe depression. Chinese version SDS was used in

**Table 1. Demographic characteristics and SDS level of medical students.**

| Characteristics | Total | Number of Non-depression (%) | Number of depressed (%) | Depression level | | | $x^2$ | P | $OR^a$(95%CI) |
| --- | --- | --- | --- | --- | --- | --- | --- | --- | --- |
| | | | | Mild (%) | Moderate (%) | Sever (%) | | | |
| **Specialty** | | | | | | | 46.75 | 0.057 | |
| Clinical medicine | 295 | 192 (65) | 103 (35) | 47 (46) | 54 (52) | 2 (2) | | | 1.00 |
| Nursing | 458 | 329 (72) | 129 (28) | 56 (43) | 68 (53) | 5 (4) | | | 0.629 (0.380,1.043) |
| Oral medicine | 174 | 132 (76) | 42 (24) | 17 (40) | 23 (55) | 2 (5) | | | 0.861 (0.528,1.404) |
| Anesthesiology | 78 | 51 (65) | 27 (35) | 11 (41) | 16 (59) | 0 (0) | | | 1.061 (0.604,1.865) |
| Preventive medicine | 146 | 99 (68) | 47 (32) | 23 (49) | 20 (43) | 4 (8) | | | 0.638 (0.335,1.215) |
| Pharmacy | 62 | 45 (73) | 17 (27) | 12 (71) | 4 (24) | 1 (6) | | | 0.711 (0.405,1.250) |
| Medical imaging | 190 | 145 (76) | 45 (24) | 21 (47) | 21 (47) | 3 (6) | | | 0.894 (0.438,1.824) |
| Rehabilitation Therapeutics | 125 | 80 (64) | 45 (36) | 17 (38) | 25 (56) | 3 (6) | | | 1.088 (0.624,1.898) |
| Medical technology | 158 | 111 (70) | 47 (30) | 26 (55) | 20 (43) | 1 (2) | | | 0.981 (0.504,1.909) |
| Biomedical engineering | 120 | 80 (67) | 40 (33) | 15 (37) | 25 (63) | 0 (0) | | | 0.649 (0.339,1.244) |
| Information management and system | 103 | 77 (75) | 26 (25) | 17 (65) | 7 (27) | 2 (8) | | | 0.675 (0.376,1.212) |
| **Grade** | | | | | | | 20.03 | 0.018 | |
| Freshman | 644 | 484 (75) | 160 (25) | 84 (53) | 71 (44) | 5 (3) | | | 1.00 |
| Sophomore | 552 | 361 (65) | 191 (35) | 83 (43) | 101 (53) | 7 (4) | | | 0.331 (0.276,0.395) |
| Junior | 329 | 227 (69) | 102 (31) | 40 (39) | 57 (56) | 5 (5) | | | 0.529 (0.444,0.631) |
| Senior | 384 | 269 (70) | 115 (30) | 55 (48) | 54 (47) | 6 (5) | | | 0.449 (0.356,0.568) |
| **Gender** | | | | | | | 28.93 | 0.001 | |
| Male | 519 | 340 (66) | 179 (34) | 65 (36) | 112 (63) | 2 (1) | | | 1.00 |
| Female | 1390 | 1001 (72) | 389 (28) | 197 (51) | 171 (44) | 21 (5) | | | 0.526 (0.439,0.631) |
| **Place of residence** | | | | | | | 2.91 | 0.820 | |
| Urban | 443 | 311 (70) | 132 (30) | 57 (43) | 71 (54) | 4 (3) | | | 1.00 |
| Rural-urban | 442 | 310 (70) | 132 (30) | 59 (45) | 65 (49) | 8 (6) | | | 1.005 (0.788,1.283) |
| Rural | 1024 | 720 (70) | 304 (30) | 146 (48) | 147 (48) | 11 (4) | | | 1.008 (0.790,1.287) |
| **One-child family** | | | | | | | 1.57 | 0.667 | |
| Yes | 396 | 282 (71) | 114 (29) | 47 (42) | 62 (54) | 5 (4) | | | 1.00 |
| No | 1516 | 1059 (70) | 454 (30) | 215 (47) | 221 (49) | 18 (4) | | | 0.943 (0.739,1.203) |
| **With good family relationship** | | | | | | | 54.12 | <0.001 | |
| Yes | 1725 | 1254 (73) | 471 (27) | 217 (46) | 240 (51) | 14 (3) | | | 1.00 |
| No | 184 | 87 (47) | 97 (53) | 45 (47) | 43 (44) | 9 (9) | | | 0.337 (0.248,0.458) |

Note: Bonferroni correction showed that there was different between freshman and sophomore in depression.

[a]: it means that the size of risk for a characteristic between the medical students with depression and no depression.

the current study. The reliability and validity of SDS have been confirmed in Liao Y's study [20]. The Cronbach's α was 0.805.

## Data analysis

The Statistical Product and Service Solutions (SPSS) software23.0 was used for statistical analysis. Some variables were normally distributed, such as age and SDS score, mean and standard deviations(SD) were reported. Frequency data, such as grade, gender and coping styles, were presented as absolute values (n,%). The relationships between the SDS levels and independent variables were analyzed with Chi-square test. The difference between groups in the percentage of reported problems was analyzed by Chi-square test with Bonferroni correction. Fisher's exact test was used when exact theory frequency was less than 1. Multivariate logistic regression analysis was conducted to explore the relationships between coping styles and independent variables. Statistical significance was set at 0.05 using two-side tests.

This study is reported as per the Strengthening the Reporting of Observational Studies in Epidemiology (STROBE) guideline (S1 File).

## Result

### Characteristics of respondents

Among the 1931 original respondents, 1909 participated in the study. The demographic and selected characteristics of the respondents were shown in Table 1. Respondents were 18–23 years, a mean age of 19.85 (SD: 1.48) years, 72.8% were female, 27.2% were male. 46.4% lived in rural areas, 20.7% came from one-child family, and 90.4% had a good family relationship. The medical students with the characteristic of sophomore ($p = 0.003$), female ($p = 0.006$), or poorly family relationship ($p<0.001$) have higher SDD scores (Table 1).

### SDS results

The medical students have mean SDS scores of 44.29 (SD = 11.67), with a range of score from 25 to 85. The prevalence of depression in the respondents can be seen in Table 2. Almost three-tenths of respondents were experiencing depression, with a mean SDS score of 59.13 (SD = 5.67), while 70.3% of medical students were not depressed, with a mean SDS score of 38.00 (SD = 6.88).

### Depression knowledge

The total correct rate for knowledge of depression among medical students was 64.14%. The question with the highest accuracy was "depression can relapse", with the correct rate of 93.2%, the second was "depression associated with hormone secretion" with correct rate of 91.3%, followed by "depression is a disease" with correct rate of 89.8%. There were statistical differences

**Table 2. Depression levels of medical students (N = 1909).**

| Depression level | Score ($\bar{x} \pm SD$) | Frequency | Ratio (%) |
|---|---|---|---|
| Normal | 38.00±6.88 | 1341 | 70.2 |
| Mild | 54.17±2.46 | 262 | 13.7 |
| Moderate | 62.47±1.95 | 283 | 14.8 |
| Severe | 74.61±4.52 | 23 | 1.2 |
| Total | 44.29±11.67 | 1909 | 100.0 |

between depressed students and non-depression students on the rate of the correct answer in the following questions: "females have more probability" "depression can be adjusted by oneself" "associated with ones character" "know treatment method of depression" "know drug use of depression" "know depression influence for health" and "know prevention method of depression." (Table 3).

## Coping strategy comparison with depression

In respondents with depression, "ask for help from psychological consultation" was the first coping way, indicated by 329 of the medical students, which was followed by "see the doctor in a hospital" with 326 respondents, while 135 respondents had no ways of coping. There were statistically significant differences in coping styles for depression among the medical students (Table 4).

**Table 3. Number of correct answer about the knowledge of depression in medical students (N = 1909).**

| Depression knowledge | Number of correct answers (%) | Number of correct answer in depressed (%) | Number of correct answer in non-depression (%) | $x^2$ | $p$ | OR(95%CI) |
|---|---|---|---|---|---|---|
| Depression is a disease | 1714(90) | 509 (30) | 1205 (70) | 0.03 | 0.871 | 1.027 (0.744,1.418) |
| Depression can be cured by oneself | 932(49) | 287 (31) | 645 (69) | 0.83 | 0.362 | 0.907 (0.746,1.104) |
| Depression can relapse | 1779(93) | 525 (30) | 1254 (70) | 0.74 | 0.391 | 0.847 (0.580,1.238) |
| Suicide rate with depression in China | 473(25) | 139 (29) | 334 (71) | 2.92 | 0.404 | 1.024 (0.815,1.286) |
| Incidence of depression in college students | 553(29) | 150 (27) | 403 (73) | 2.58 | 0.109 | 1.197 (0.961,1.492) |
| Depression can be adjusted by oneself | 1621(85) | 453 (28) | 1168 (72) | 16.81 | <0.001 | 1.714 (1.322,2.222) |
| Associated with genetic factors | 823(43) | 209 (25) | 614 (75) | 0.10 | 0.755 | 1.034 (0.839,1.274) |
| Female have more probability | 1605(84) | 385 (24) | 1220 (76) | 5.94 | 0.015 | 1.397 (1.067,1.829) |
| Associated with hormone secretion | 1742(91) | 440 (25) | 1302 (75) | 0.51 | 0.476 | 1.147 (0.786,1.673) |
| Associated with chronic disease | 1515(79) | 382 (30) | 1133 (75) | 0.12 | 0.729 | 1.047 (0.809,1.354) |
| Associated with drug abuse | 1643(86) | 491 (24) | 1152 (70) | 0.10 | 0.756 | 1.014 (0.752,1.369) |
| Associated with one's personality | 1728(91) | 418 (31) | 1310 (76) | 7.01 | 0.008 | 0.643 (0.463.0.894) |
| Know symptom of depression | 1276(67) | 398 (31) | 878 (69) | 3.80 | 0.051 | 0.960 (0.905,1.017) |
| Know treatment method of depression | 1431(75) | 366 (26) | 1065 (74) | 47.71 | <0.001 | 1.150 (1.080,1.225) |
| Know drug use of depression | 462(24) | 157 (34) | 305 (66) | 5.22 | 0.022 | 0.711 (0.563,0.898) |
| Know side-effect of depression cure | 1168(61) | 340 (29) | 828 (71) | 0.59 | 0.440 | 1.085 (0.979,1.203) |
| Know depression influence for health | 1530(80) | 438 (29) | 1092 (71) | 21.62 | <0.001 | 1.365 (1.154,1.616) |
| Know prevention method of depression | 1660(87) | 386 (23) | 1274 (77) | 21.64 | <0.001 | 0.517 (0.390,0.681) |

**Table 4. Coping strategy comparison with depression (n = 568).**

| Coping way | with depression | mild (%) | moderate (%) | sever (%) | $x^2$ | $p$ |
|---|---|---|---|---|---|---|
| No ways to coping | 135(24) | 42(31) | 83(62) | 10(7) | 18.42 | <0.001 |
| Ask friends for help | 294(52) | 138(47) | 151(51) | 5(2) | 8.68 | 0.013 |
| Search a book | 272(48) | 127(47) | 140(51) | 5(2) | 6.62 | 0.036 |
| Ask for help from psychological consultation in the college | 329(58) | 156(47) | 165(50) | 8(3) | 5.35 | 0.069 |
| See the doctor in a hospital | 326(57) | 159(49) | 157(48) | 10(3) | 3.41 | 0.182 |

## Logistic regression analysis

Only the medical students with depression conducted the Logistic regression analysis. Coping styles for depression were the dependent variable. The coping ways of "ask friends for help" "search a book" "ask for help from psychological consultation in the college and "see the doctor in a hospital" were defined as "coping with", while the coping way of "no ways of coping " was defined as "no way of coping". Then, coping styles were used as dependent variables, and gender, age, specialty, grade, place of residence, one-child family, with good family relationship, and depression knowledge score were classed as independent variables. Logistic regression models were conducted in medical students with depression. Female OR = 1.470 (1.078, 2.005), residence in rural area OR = 1.496 (1.038, 2.156), poor family relationship OR = 2.428 (1.790, 3.293), and lower cognitive level of depression OR = 1.920 (1.426, 3.226) showed significant relationship with "no ways of coping" (Table 5).

## Discussion

The rate of depression in medical students in our study was 29.7%, with a mean SDS score of 59.13±5.67. The prevalence rate was similar to a prospective, longitudinal observational study conducted in Portugal which the depression rate ranged from 21.5 to 12.7% [21]. The results revealed that medical students in China have a similar prevalence compared with many studies. As noted by many authors about depression of medical students [11, 22, 23], there are several possible reasons why medical students in China obtained a slightly high depression score. Firstly, medical students spend a long time in college and hospital in China, generally 5 years or 8 years, or longer, which is a stressful and competitive period. Secondly, medical students are overburdened. They must memorize a large amount of information in a limited time.

**Table 5. Logistic regression analysis results between coping styles and influence factors.**

| Influence factors | | B | SE | Wald | p | OR(95%CI) |
|---|---|---|---|---|---|---|
| Gender | Male | | | | | 1 |
| | Female | 0.385 | 0.158 | 0.015 | | 1.470 (1.078~2.005) |
| Place of residence | Urban | | | | | 1 |
| | Rural-urban | 0.274 | 0.212 | 0.195 | | 1.316 (0.868~1.994) |
| | Rural | 0.403 | 0.186 | 0.031 | | 1.496 (1.038~2.156) |
| With good family relationship | Yes | | | | | 1 |
| | No | 0.763 | 0.208 | <0.001 | | 2.415 (1.426~3.226) |
| Knowledge level of depression | High | | | | | 1 |
| | Low | 0.652 | 0.151 | <0.001 | | 1.920 (1.427~2.583) |

* Only variables that exerted a significant relationship with coping styles were shown.

Information overload may create feelings of disappointment, and may result in an inability to handle all the information and increased errors, which ultimately breaks the stability of the student wellness. Thirdly, medical students still experience immense pressure, such as the stress of the long length of schooling, academic pressure, the stress of clinical practice, etc. Fourthly, the information of our study got in the period of COVID-19 pandemic, medical staffs might suffer from great mental health problems, many studies showed already COVID-19 pandemic increased the rate of depression in medical staffs [24, 25], which might effect indirectly health of medical student. Therefore, it was important to provide psychological interventions for medical student to reduce their depression in the period of COVID-19 pandemic. At the same time, targeted mental health education should been conducted in daily work, and to find psychological problem in timely.

The present study indicated that female medical students had a higher risk of suffering from depression compared with male medical students. Different studies have reported mixed findings on comparing depression by gender among medical students. Lloyd found that female medical students from the first to fourth years students scored higher on the depression scales compared with male students [26]. Clark observed no difference in the probability of becoming depressed based on gender in a four-year longitudinal study [27]. In a Meta-analysis of diagnoses and symptoms of depression in over 90 different nations, gender differences for diagnoses emerged at the age of 12 years [28].

In the current study, poor family relationship was associated with a higher SDS score. Family relationships play an important role in providing psychological support to Chinese medical students. Physical and mental well-being of medical students may be improved by emotional comforts, love, and warmth of the family function. Numerous studies among medical students have demonstrated similar results. For example, high levels of family conflict and low levels of family cohesiveness were associated with depression [29]. Depression had highly significant correlations with family functioning among medical students in China [30]. Chen et al. reported that there was significantly negative correlation between parent-child relationship, family functioning and depression in college students [31]. A study by Thompson G found that a greater risk of depression among medical students was associated with inadequate support from family [32], at the same time, Moon and Rao reported that youth-family relationships were significant predictors of adolescent depression [33]. Yamada R et al. reported that depression populations with adverse childhood emotion experiences had significantly higher score [34].

Personal knowledge of depression influenced the recognition and management of symptoms, help-seeking patterns, and the use of services. In 1997 [35], Jorm et al. proposed the concept of "mental health literacy" and defined it as "knowledge and beliefs about mental disorders which aid their recognition, management or prevention" Mental health literacy includes six components: the ability to identify specific disorders or psychological distress; knowledge and beliefs about risk factors and causes; self-help interventions; professional help available; how to seek mental health information; and attitudes that facilitate recognition and appropriate help-seeking [36]. A person with sufficient knowledge is more likely to obtain more health knowledge and will be able to identify the mental disorder, and hence will be more proactively seek effective treatment. Jorm et al. found that knowledge and beliefs about mental health could affected help-seeking patterns and use of services [35].

In our studies, 23.8% of medical students with depression classed as no ways of coping. The result was similar to Thompson G's study that 23% responded unable to cope [32]. On the one hand, some medical students considered that treatment would not help on depression. On the other hand, they were afraid of being perceived as mental disease [11]. In our study, "ask for help from psychological consultation in the college" was selected as the most frequent coping

way for depressed medical student. Fisher and Goldney found that physicians were the most useful style for college students in the case of depression [37]. In a study in Australia in which participants were asked to rank sources of help for depression, the first source was counselors, followed by close friends, next to general practitioners and psychologists [38]. Thompson G et al. reported that medical students frequently used approach-oriented coping strategies, such as contacting a therapist or counselor, talking with friends or family members, and seeking support from a church or spiritual advisor [32]. Our study result were similar to previous studies. A study in Germany in 1999 found that professionals were the most frequently used helpful source for depression, followed by mental health professionals, and general health providers [39]. It suggests that some medical students know the correct ways to cope with depression, but some of them ignore depressive symptom and consequently do not seek help. Therefore, it is necessary for medical students to improve their knowledge and beliefs about depression risk factors. Medical colleges have advantages in education professionals and equipment, targeted health education can help medical student to adjust to social isolation, get social support, acquire abilities to effectively cope during unprecedented times.

## Limitation

There were limitations to this study. Firstly, online-based self-reporting survey has certain limitations compared with face-to-face interviews, the reality of information cannot be identified. Therefore, extrapolation of the results should be cautious. Secondly, medical students in our study were not be selected randomly, the study was conducted only in a city in the southeast of Shanxi province in China, so, they didn't represent the entire medical student population in China, the depression levels and coping styles may diverse accompanied by cultural differences, speed of social development, and other factors. Therefore, it is necessary to assess medical students in medical colleges in other areas of China in the future.

## Conclusion

In summary, this study has some important practical implications. First, it is necessary to focus on mental health of medical students, especially in female, residence in rural areas, poor family relationships, and lower cognitive level of depression knowledge. Secondly, medical students were insufficient on depression knowledge and coping styles, and efforts that educate students know risk of impaired mental health could improve diagnosis and treatment.

## Supporting information

**S1 File. STROBE checklist cross-sectional.**
(DOCX)

**S2 File. Questionnaire-in English.**
(DOCX)

**S3 File. Data.**
(XLSX)

## Author Contributions

**Conceptualization:** Yajun Lian, Weiwei Ping.

**Data curation:** Yumeng Yan.

**Formal analysis:** Yumeng Yan.

**Funding acquisition:** Weiwei Ping.

**Investigation:** Yumeng Yan, Zhiyong Dou, Xiaoyan Wang, Hui Yang.

**Methodology:** Yajun Lian, Yumeng Yan.

**Project administration:** Weiwei Ping.

**Resources:** Weiwei Ping.

**Software:** Yumeng Yan.

**Writing – original draft:** Yajun Lian.

**Writing – review & editing:** Yumeng Yan.

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
