## [Decision Letter · Decision Letter 0]

18 Aug 2022

PONE-D-22-10652Knowledge and Coping Style About Depression in Medical Students : A Cross-Sectional Study in North ChinaPLOS ONE

Dear Dr. Ping,

Thank you for submitting your manuscript to PLOS ONE. After careful consideration, we feel that it has merit but does not fully meet PLOS ONE’s publication criteria as it currently stands. Therefore, we invite you to submit a revised version of the manuscript that addresses the points raised during the review process.

Two external reviewers have commented on your manuscript and have identified a range of issues that need to be carefully addressed in a revision. Please pay particular attention to responding to their concerns regarding the clarity of the study objectives and the methodological details. 

We look forward to receiving your revised manuscript.

Kind regards,

Jamie Males

Editorial Office

PLOS ONE

Journal Requirements:

5. We note you have included a table to which you do not refer in the text of your manuscript. Please ensure that you refer to Table 5 in your text; if accepted, production will need this reference to link the reader to the Table.

7.  Thank you for submitting the above manuscript to PLOS ONE. During our internal evaluation of the manuscript, we found significant text overlap between your submission and the following previously published works, some of which you may be an author:

- https://journals.sagepub.com/doi/10.1177/0733464810374468

- https://www.cambridge.org/core/journals/international-psychogeriatrics/article/abs/lay-perceptions-about-mental-health-where-is-age-and-where-is-alzheimers-disease/08DD8B5191E8B6897AB3C24DA4E26BF7

- https://amhsjournal.org/article.asp?aulast=Bose&epage=235&issn=2321-4848&issue=2&spage=230&volume=8&year=2020

- https://www.semanticscholar.org/paper/Systematic-Review-of-Depression%2C-Anxiety%2C-and-Other-Dyrbye-Thomas/32c0691e8abd154312d5555193d0ff9fe224e5c4?p2df

- https://www.who.int/news-room/fact-sheets/detail/depression

Please revise the manuscript to rephrase the duplicated text, cite your sources, and provide details as to how the current manuscript advances on previous work. Please note that further consideration is dependent on the submission of a manuscript that addresses these concerns about the overlap in text with published work.

Reviewers' comments:

Reviewer's Responses to Questions

**Comments to the Author**

1. Is the manuscript technically sound, and do the data support the conclusions?

Reviewer #1: Yes

Reviewer #2: Partly

2. Has the statistical analysis been performed appropriately and rigorously? 

Reviewer #1: Yes

Reviewer #2: Yes

3. Have the authors made all data underlying the findings in their manuscript fully available?

Reviewer #1: Yes

Reviewer #2: Yes

4. Is the manuscript presented in an intelligible fashion and written in standard English?

Reviewer #1: No

Reviewer #2: No

5. Review Comments to the Author

Reviewer #1: General remarks

From the summary to the discussion, there are typographical errors, due to lack of spaces. For example, see lines: 40, 45, 50, 105, 106, 124, 136, 151, 158, 177, 179, 180, 181, 192, 200, 218, 222, 243, 245, 246, 247, 249, 251, 255, 256, 257, 287, 288.

On line 182 it says "SDD”, and I guess it should say "SDS".

Line 183, Table 2, say “meidcal”

The citations, particularly noted in the discussion, do not correspond to the references. For example, 35 says Liselotte and is Mao, 37 says Lloyd and is Camp, 38 says Camp and is Clark, 39 says Clark and is Chen, …

Abstract

The results in terms of students' knowledge about depression are not observed

Introduction

1) The objectives of the study are not clear; they seem to be only descriptive but in their results, they present data of associations that are not indicated as objectives.

2) It is missing that they indicate the hypotheses (and their foundation) of the main associations of the study. For example, the authors could point out how knowledge affects the type of coping, or how knowledge and coping are associated with the presence of depression (depending on the real objectives of the study).

Methods

Indicate the total number of students invited to participate and the response rate.

It is not clear how the authors evaluated coping: by one or several questions? was/were open-ended or students could choose from multiple options?

In the statistical analysis, the authors indicate that they will present the percentage of people in each group (line 155), but it is not clear to which groups they refer.

Results

In point 3 of the results (line 186 to 192), the information that was already indicated in the method could be omitted.

Tables

1) Check that the sentences begin with a capital letter.

2) The tables do not have a presentation order, that is, they present Table 2 first.

3) In Table 2, the word "frequency" could be used instead of "number".

4) The tables do not specify which values are in parentheses. See Table 1 (columns 2 and 3), Tables 3 and 4.

Discussion

Missing put a paragraph with the conclusions.

Check references.

Reviewer #2: Overall, this is an interesting study that looked at depression through the lens of mental health literacy within a sample of Chinese medical students. The sample size was more than ample to detect effects. However, there were a lot of issues in the quality of communication that significantly impeded accessibility of the study. The manuscript would benefit from language editing. It was not practical to identify all the language issues within the manuscript as they were prevalent and significant. Some specific points for improvement:-

Introduction

-The intro overly focuses on presenting depression statistics and prevalence rates. However, this needs further development in order to effectively build a line of argument. For example, the authors could explore and expand on factors linked/related to the observed increased prevalence rates in college/university students as the population of interest for the current study.

-More is needed to contextualise the study in the geographical context within which the study was conducted. It needs to be clear what the state of knowledge is around depression within China, and in particular among Chinese university students.

-The literature on treatments for depression is presented in an entirely descriptive manner, and it is not directly relevant to the present study.

-The rationale for the present study is not clear. The introduction has not managed to make a case for why this particular study is needed, in this particular context, and within this particular population. It is not clear what the gap in knowledge is, that the present study aims to fill.

-Lines 98-100 need clarifying. It is not clear what the conflict being referred to here is. It is also not clear how the present study aims to resolve this conflict. The point about distinguishing depression in the population lacks clarity.

-Lines 100-103 need to be further expanded in order to help develop a rationale for the present study. In what way is gathering this information beneficial? The authors need to explicitly state how knowledge of the role of these factors would be beneficial in advancing the knowledge base further.

Methods

-A summary of demographic information for participants should be reported under a ‘Participants’ sub-heading (not ‘respondents’).

-The knowledge and coping style measure (lines 123-130) needs to be described more clearly. Including a couple of sample items will be helpful. It is unclear whether ‘knowledge’ and coping style are two separate subscales. As this is a newly developed measure, internal consistency and reliability/validity analyses should be reported. The scoring criteria is not described in an accessible manner.

Results

-Lines 180-182- reports factors ‘associated’ with higher SDS score. This needs revising for clarity, as no tests of association were conducted

-Lines 186-200- the description of the scoring for the depression knowledge measure should be clarified and placed in the ‘Method’. It is also not clear why analyses were conducted for individual items of the measure.

-Lines 217-229- the results reported here are inaccessible. It is not clear what the figures are referring to.

-Lines 224- 237- logistic regression results are difficult to decipher.

-Table 5 has not been referenced within the text.

Discussion

- Lines242-261- provide comparisons in prevalence rates between present and prior studies. It is not clear whether the prior studies were conducted in medical students specifically. And if this is the case, lines 251-261 are redundant as the reasons for high prevalence rates presented here are applicable to all medical students. The authors need to specifically explore reasons for differences in prevalence that are unique to Chinese medical students as compared to medical students from other countries.

-Lines 294-296 & Lines 303-306 need clarifying

-Limitations need to be much more substantive. For example, it is not clear how utilising a single measure of depression was a limitation. It is also not clear what the limitations of self report alluded to are.

6. PLOS authors have the option to publish the peer review history of their article (what does this mean?). If published, this will include your full peer review and any attached files.

Reviewer #1: No

Reviewer #2: No

---

## [Author Response · Author response to Decision Letter 0]

13 Oct 2022

Our responds to the editor’s and the reviewer’s comments：

Reply to editor

We thank you very much for the constructive comments and suggestions on our manuscript. We have corrected our manuscript according to the comments.

Reply: We accepted your suggestion, and download the PLOS ONE style templates from the website as you noted. We revised manuscript to meets PLOS ONE's style requirements.

Reply: we provided additional details regarding participant consent in our revised manuscript and supporting information files (S1).

Reply: we agreed your suggestion

Reply: we agreed your suggestion and deleted ethics statement appeared in the other section

5. We note you have included a table to which you do not refer in the text of your manuscript. Please ensure that you refer to Table 5 in your text; if accepted, production will need this reference to link the reader to the Table.

Reply: we thank very much for your suggest on Table 5. We confirmed the text of the Table 5 in revised manuscript.

6.Please include captions for your Supporting Information files at the end of your manuscript, and update any in-text citations to match accordingly. Please see our Supporting Information guidelines for more information: http://journals.plos.org/plosone/s/supporting-information.

Reply: After careful saw Supporting Information guidelines, we added captions for our Supporting Information files at the end of manuscript.

7.  During our internal evaluation of the manuscript, we found significant text overlap between your submission and the following previously published works, some of which you may be an author: 

 https://journals.sagepub.com/doi/10.1177/0733464810374468

- https://www.cambridge.org/core/journals/international-psychogeriatrics/article/abs/lay-perceptions-about-mental-health-where-is-age-and-where-is-alzheimers-disease/08DD8B5191E8B6897AB3C24DA4E26BF7

- https://amhsjournal.org/article.asp?aulast=Bose&epage=235&issn=2321-4848&issue=2&spage=230&volume=8&year=2020

- https://www.semanticscholar.org/paper/Systematic-Review-of-Depression%2C-Anxiety%2C-and-Other-Dyrbye-Thomas/32c0691e8abd154312d5555193d0ff9fe224e5c4?p2df

- https://www.who.int/news-room/fact-sheets/detail/depression

Please revise the manuscript to rephrase the duplicated text, cite your sources, and provide details as to how the current manuscript advances on previous work. Please note that further consideration is dependent on the submission of a manuscript that addresses these concerns about the overlap in text with published work.

Reply: We thank you very much for the comments and suggestions on our manuscript. We invied the Charlesworth Author services to polish the manuscript. After that, we checked the manuscript word-for-word to ensure the quality .

Reply to reviewer1 

We thank you very much for the constructive comments and suggestions on our manuscript. We have corrected our manuscript according to the comments.

 1.From the summary to the discussion, there are typographical errors, due to lack of spaces. For example, see lines: 40, 45, 50, 105, 106, 124, 136, 151, 158, 177, 179, 180, 181, 192, 200, 218, 222, 243, 245, 246, 247, 249, 251, 255, 256, 257, 287, 288.

On line 182 it says "SDD”, and I guess it should say "SDS".

Line 183, Table 2, say “meidcal”

The citations, particularly noted in the discussion, do not correspond to the references. For example, 35 says Liselotte and is Mao, 37 says Lloyd and is Camp, 38 says Camp and is Clark, 39 says Clark and is Chen, …

Reply: We thank you very much for the comments and suggestions on our manuscript. We checked and corrected the manuscript word-for-word to ensure the quality. The errors noted above have been corrected, and showed in the manuscrip.

2.Abstract: The results in terms of students' knowledge about depression are not observed

Reply: we added text related the results in terms of students' knowledge about depression in abstract section, and description showed in the manuscript as follows: There were statistical differences between the with depression students and the no depression students in question: “Females has more probability”, “Depression can be adjusted by oneself,” “Associated with one`s character,” “Know cure method of depression,” “Know drug use of depression,” “Know depression influence for health,” and “Know prevention method of depression.” 

3.Introduction

1) The objectives of the study are not clear; they seem to be only descriptive but in their results, they present data of associations that are not indicated as objectives.

2) It is missing that they indicate the hypotheses (and their foundation) of the main associations of the study. For example, the authors could point out how knowledge affects the type of coping, or how knowledge and coping are associated with the presence of depression (depending on the real objectives of the study).

Reply: We agree with this suggestion and have modified the throughout introduction section as appropriate. the text has been showed in manuscript.

4.Methods

Indicate the total number of students invited to participate and the response rate.

Reply: we agree and accept your suggestion, the description about “the total number of students invited to participate and the response rate” has been added as following “During the period of work, 2200 medical students were invited. By March 20, a total of 1931 questionnaires had been returned by respondents, and the response rate was 87.8%.”

It is not clear how the authors evaluated coping: by one or several questions? was/were open-ended or students could choose from multiple options?

Reply: we agree and accept your suggestion, the description about coping styles has been added as follows “The copying strategy was assessed by the question, “if you are affected by depression, which copying style would be chosen (multiple choice) ?” “No coping way,” “ask for help from friends,” “ search a book, ask for help from psychological consultation office in the college,” “see the doctor to hospital” had been listed choices.

In the statistical analysis, the authors indicate that they will present the percentage of people in each group (line 155), but it is not clear to which groups they refer.

Reply: we agree and accept your suggestion. After carefully consideration, we deleted the percentage of people in each group (line 155) .

5.Results

In point 3 of the results (line 186 to 192), the information that was already indicated in the method could be omitted.

Reply: we agree and accept your suggestion. We deleted the content you mentioned.

Tables

1)Check that the sentences begin with a capital letter.

Reply:we agree and accept your suggestion.We check our text and use capital letter in the right place.

2)The tables do not have a presentation order, that is, they present Table 2 first.

Reply:we agree and accept your suggestion. We relist all tables with correctly order.

3)In Table 2, the word "frequency" could be used instead of "number".

Reply:we agree and accept your suggestion. We use the word "frequency"in Table 2.

4)The tables do not specify which values are in parentheses. See Table 1 (columns 2 and 3), Tables 3 and 4.

Reply:we agree and accept your suggestion. After carefully consideration, we deleted the percentage of people in Table 1 (columns 2 and 3), Tables 3 and 4.

5.Discussion

Missing put a paragraph with the conclusions.

Reply:we agree and accept your suggestion. We add the conclusion paragraph in our manuscript.

6.Check references.

Reply:we agree and accept your suggestion. After carefully revision, we relist and check all references in our manuscript.

Reply to reviewer2 

We thank you very much for the constructive comments and suggestions on our manuscript. We have corrected our manuscript according to your comments.

1.Introduction

-The intro overly focuses on presenting depression statistics and prevalence rates. However, this needs further development in order to effectively build a line of argument. For example, the authors could explore and expand on factors linked/related to the observed increased prevalence rates in college/university students as the population of interest for the current study.

Reply: We agree with this suggestion and have modified the throughout introduction section as appropriate. the text has been showed in manuscript.

-More is needed to contextualise the study in the geographical context within which the study was conducted. It needs to be clear what the state of knowledge is around depression within China, and in particular among Chinese university students.

Reply: We agree with this suggestion and have modified the throughout introduction section as appropriate. the text has been showed in manuscript.

-The literature on treatments for depression is presented in an entirely descriptive manner, and it is not directly relevant to the present study

Reply: we agree with this suggestion. We deleted the content about treatments for depression.

-The rationale for the present study is not clear. The introduction has not managed to make a case for why this particular study is needed, in this particular context, and within this particular population. It is not clear what the gap in knowledge is, that the present study aims to fill.

The current study aimed to ascertain medical students` depression situation and their levels of knowledge about depression, then subsequently explore the relationship between level of knowledge and coping style of the medical students on depression. 

-Lines 98-100 need clarifying. It is not clear what the conflict being referred to here is. It is also not clear how the present study aims to resolve this conflict. The point about distinguishing depression in the population lacks clarity.

-Lines 100-103 need to be further expanded in order to help develop a rationale for the present study. In what way is gathering this information beneficial? The authors need to explicitly state how knowledge of the role of these factors would be beneficial in advancing the knowledge base further.

Reply:we agree with 4 suggestion you noted. After carefully revision, the rationale for the present study were represent as follows “ Therefore, we found a conflict between the high rate of depression and the low rate of treatment. The key point is to distinguish depression in the population. To date, limited attention has been directed at individuals` knowledge, beliefs, and ways of coping with depression. Exploration of such knowledge and coping style(s) for depression may reveal factors related to the conflict. ”

2.Methods

-A summary of demographic information for participants should be reported under a ‘Participants’ sub-heading (not ‘respondents’).

Reply:we agree and accept your suggestion. We use the word "Participants"in the part of summary of demographic information for participants.

-The knowledge and coping style measure (lines 123-130) needs to be described more clearly. Including a couple of sample items will be helpful. It is unclear whether ‘knowledge’ and coping style are two separate subscales. As this is a newly developed measure, internal consistency and reliability/validity analyses should be reported. The scoring criteria is not described in an accessible manner.

Reply: we agree and accept your suggestion, the description about coping styles has been added as following “The copying strategy was assessed by the question, “if you are affected by depression, which copying style would be chosen (multiple choice) ?” “No coping way,” “ask for help from friends,” “ search a book, ask for help from psychological consultation office in the college,” “see the doctor to hospital” had been listed choices. 

 The part “knowledge about depression” was only a questionnaire designed by ourself,not a scale, so,we did not assess internal consistency and reliability/validity

3.Results

-Lines 180-182- reports factors ‘associated’ with higher SDS score. This needs revising for clarity, as no tests of association were conducted

Reply:we agree and accept your suggestion. The description you mentioned has been corrected as follows “ The medical students with the characteristic of sophomore (p=0.003), female (p=0.006), and poor family relationship (p<0.001) have higher SDD scores”

-Lines 186-200- the description of the scoring for the depression knowledge measure should be clarified and placed in the ‘Method’. It is also not clear why analyses were conducted for individual items of the measure.

Reply: we agree and accept your suggestion.We deleted the content you mentioned. The aims to analysis individual items was to found which knowledges were short for medical students.

-Lines 217-229- the results reported here are inaccessible. It is not clear what the figures are referring to.

-Lines 224- 237- logistic regression results are difficult to decipher.

-Table 5 has not been referenced within the text.

Reply: we agree and accept your suggestion.We show the results you mentioned by Table5. We corrected the text of logistic regression results in our manuscript as follows “Coping styles for depression were dichotomous. The ways of “ask for help from friends,” “search a book,” “ask for help from psychological consultation office in the college,” and “see the doctor to hospital” were defined as “coping with,” while “no coping way” was defined as “no to coping with.” Then, coping styles were used as dependent variables, and gender, age, specialty, grade, place of residence, one-child family, with good family relationship, and depression knowledge score were classed as independent variables. Multivariate logistic regression models were conducted in medical students with depression. Female (OR=1.470, 95%CI: 1.078, 2.005), residence in rural area (OR=1.496, 95%CI: 1.038, 2.156), poor family relationship (OR=2.428, 95%CI: 1.790, 3.293), and lower cognitive level of depression (OR=1.920, 95%CI: 1.426, 3.226) showed significantly relationship with “no to coping with” (Table 5).”

Discussion

- Lines242-261- provide comparisons in prevalence rates between present and prior studies. It is not clear whether the prior studies were conducted in medical students specifically. And if this is the case, lines 251-261 are redundant as the reasons for high prevalence rates presented here are applicable to all medical students. The authors need to specifically explore reasons for differences in prevalence that are unique to Chinese medical students as compared to medical students from other countries.

Reply: Studies mentioned in lines 242-261 were conducted in medical students specifically, so, the reasons were similar to other countries.

-Lines 294-296 & Lines 303-306 need clarifying

Reply: we agree and accept your suggestions. We modify the parts you mentioned in manuscript.

-Limitations need to be much more substantive. For example, it is not clear how utilising a single measure of depression was a limitation. It is also not clear what the limitations of self report alluded to are.

Reply: we agree and accept your suggestions. We modify the part of limitations according your suggestions.

Once again, thank you very much for your constructive comments and suggestions which would help us both in English and in depth to improve the quality of the paper.

---

## [Decision Letter · Decision Letter 1]

5 Dec 2022

PONE-D-22-10652R1Knowledge and Coping Style About Depression in Medical Students : A Cross-Sectional Study in North ChinaPLOS ONE

Dear Dr. Ping,

Thank you for submitting your manuscript to PLOS ONE. After careful consideration, we feel that it has merit but does not fully meet PLOS ONE’s publication criteria as it currently stands. Therefore, we invite you to submit a revised version of the manuscript that addresses the points raised during the review process.

We look forward to receiving your revised manuscript.

Kind regards,

Callam Davidson

Editorial Office

PLOS ONE

Journal Requirements:

Please provide version of the items in the Supporting Information translated into English to allow for editorial review (Column headers in the ‘Data’ file and the ‘Questionnaire’).

Please ensure items in the Supporting Information are named using PLOS ONE conventions. The following guidance should be followed: https://journals.plos.org/plosone/s/supporting-information

*PLOS ONE* does not copyedit accepted manuscripts, so the language in submitted articles must be clear, correct, and unambiguous. We recommend that authors seek independent editorial help before submitting a revision. These services can be found on the web using search terms like “scientific editing service” or “manuscript editing service.”

Please ensure that the study is reported according to the STROBE guideline, and include the completed STROBE checklist as Supporting Information.  Please add the following statement, or similar, to the Methods: "This study is reported as per the Strengthening the Reporting of Observational Studies in Epidemiology (STROBE) guideline (S1 Checklist)."

Your Financial Disclosure from the submission form does not match the ‘Funding’ section at the end of the manuscript. Please ensure both are correct.

Reviewers' comments:

Reviewer's Responses to Questions

**Comments to the Author**

1. If the authors have adequately addressed your comments raised in a previous round of review and you feel that this manuscript is now acceptable for publication, you may indicate that here to bypass the “Comments to the Author” section, enter your conflict of interest statement in the “Confidential to Editor” section, and submit your "Accept" recommendation.

Reviewer #1: All comments have been addressed

Reviewer #2: (No Response)

2. Is the manuscript technically sound, and do the data support the conclusions?

Reviewer #1: Yes

Reviewer #2: Yes

3. Has the statistical analysis been performed appropriately and rigorously? 

Reviewer #1: Yes

Reviewer #2: No

4. Have the authors made all data underlying the findings in their manuscript fully available?

Reviewer #1: Yes

Reviewer #2: Yes

5. Is the manuscript presented in an intelligible fashion and written in standard English?

Reviewer #1: Yes

Reviewer #2: No

6. Review Comments to the Author

Reviewer #1: The authors made all the changes, but I think some were misunderstood, for example removing the percentages and notes from the tables. Unless someone else has requested it.

The observations are already very brief, above all, to the presentation of the tables:

Table 1

I do not understand why the authors decided to eliminate the percentages, I hope my observation has not been misunderstood. In case of putting the percentages again, it would be advisable to put them without decimals.

It would be advisable to leave the values of the statistics (chi square and OR [but not the “p” values]) with two decimal places.

From my point of view, the standards give clarity to the table. Why remove them, did any reviewers suggest it?

Table 3

Be more detailed in the title, i.e. point out "what is associated with what" and/or be more specific in the table footnote.

Put percentages without decimals. Take care that it is all the data that shows a frequency.

Present the values of the statistics (X2 and OR) only with two decimal places.

Also, a typographical error check is again recommended, as in "sofeware" (Line 148)

Reviewer #2: Thank you for a considered attempt at addressing my earlier comments. However, there is still a significant number of issues that require addressing as noted below.

Abstract

• Line 43-48- the observed differences between groups (with and without depression need clarifying).

• Lines 45-53- it is not clear what the dependent variable being predicted here is

• The conclusion is not clear, and none of the statements made here are accessible. For example, does the reference to ‘broadcasting mental health’ infer the importance of sharing whether one has mental health problems? It is not clear.

Introduction

• Typos- line 70 page 3- should be ‘depression’ not depressing; line 86 page 4 ‘made no treated history’- should this be ‘had no history of treatment’?

• The introduction is very brief, and lacking in analysis of the presented information. For example, page 3 line 78 could comment on and explore reasons for the significant differences in prevalence rates (6-66.5%).

Method

• Page 4 line 104- should read ‘the questionnaire survey’ or simply ‘the survey’

• Arrangements for informed consent are slightly concerning. It is not clear how consent was informed based only on a ‘statement’ particularly given the sensitive nature of the study. Did participants explicitly consent (e.g. by ticking a box) or was consent inferred by progressing with questionnaire? How much information about the study did they receive prior to completing the questionnaires?

• Page 6 line 148 typo- should be ‘software’

Results

• Table 3 needs clarifying. For example, the % of correct answers column- percentages are in brackets but not clear what the number appearing before this is? Is this number of correct answers? Does the ‘with depression’ and ‘no depression’ colums show the number in each group endorsing a correct answer? This is not clear. Finally was the chi square comparing absolute numbers in each group (depression v no depression) endorsing the correct answer or the proportions? This needs explaining in the main body. It also needs to be clear what the significant results show.

• Logistic regression results- the concept of ‘no to coping with’ which is the outcome variable needs clarifying as it is not accessible- i.e. not clear what it means.

• Multiple comparisons increase risk of family-wise error rate. Thus Bonferroni correction is needed for all analyses.

Discussion

• Page 13 line 224-225 is directly contradicting lines 220-222.

• Lines 225-227- not clear here whether the comparisons being discussed are of other medical students.

• Line 232- do you mean ‘wellness’

• Linbe 268- do you mean ‘will be qualified’?

• Line 270- do you mean ‘could affect’

• Line 271-272 (and throughout the paper)- you refer to ‘no coping way’- do you mean ‘no coping mechanisms’? or ‘no ways of coping’?

• Iine 272-273- it is not clear how your results compare to Thompson as there is a lack of clarity in regards to your conceptualisation of ‘no coping way’.

• Lines 273-281- need revising for grammar

• Lines 292-296- not substantive. Multiple tools can still capture depression adequately, and there is never an expectation for a study to use all available tools

• Lines 296-298- need revising for meaning

7. PLOS authors have the option to publish the peer review history of their article (what does this mean?). If published, this will include your full peer review and any attached files.

Reviewer #1: No

Reviewer #2: No

---

## [Author Response · Author response to Decision Letter 1]

15 Dec 2022

Our responds to the editor’s and the reviewer’s comments：

Reply to editor

We thank you very much for the constructive comments and suggestions on our manuscript. We have corrected our manuscript according to the comments.

1.Please provide version of the items in the Supporting Information translated into English to allow for editorial review (Column headers in the ‘Data’ file and the ‘Questionnaire’).

Reply: we translated the items in the Supporting Information into English and can find in the Supporting Information.We named the Supporting Information using PLOS ONE conventions.

2.PLOS ONE does not copyedit accepted manuscripts, so the language in submitted articles must be clear, correct, and unambiguous. We recommend that authors seek independent editorial help before submitting a revision. These services can be found on the web using search terms like “scientific editing service” or “manuscript editing service.”

Reply: We thank you very much for the comments and suggestions on our manuscript. We invited the friend coming from University Of Glasgow to polish the manuscript. After that, we checked the manuscript word-for-word to ensure the quality .

3.Please ensure that the study is reported according to the STROBE guideline, and include the completed STROBE checklist as Supporting Information.  Please add the following statement, or similar, to the Methods: "This study is reported as per the Strengthening the Reporting of Observational Studies in Epidemiology (STROBE) guideline (S1 Checklist)."

Reply: we reported the completed STROBE checklist as Supporting Information and add the following statement to the methods: "This study is reported as per the Strengthening the Reporting of Observational Studies in Epidemiology (STROBE) guideline (S1 Checklist)."

4.Your Financial Disclosure from the submission form does not match the ‘Funding’ section at the end of the manuscript. Please ensure both are correct.

Reply: we revised the Financial Disclosure at the end of the manuscript.

Reply to reviewer1 

We thank you very much for the constructive comments and suggestions on our manuscript. We have corrected our manuscript according to the comments.

1.Table 1

I do not understand why the authors decided to eliminate the percentages, I hope my observation has not been misunderstood. In case of putting the percentages again, it would be advisable to put them without decimals.

It would be advisable to leave the values of the statistics (chi square and OR [but not the “p” values]) with two decimal places.

From my point of view, the standards give clarity to the table. Why remove them, did any reviewers suggest it?

Reply: We thank you very much for the comments and suggestions on our manuscript. We checked and corrected the Table 1. We added the percentages without decimals and left the values of the statistics with two decimal places in all the tables in our manuscript.

2.Table 3

Be more detailed in the title, i.e. point out "what is associated with what" and/or be more specific in the table footnote.

Put percentages without decimals. Take care that it is all the data that shows a frequency.

Present the values of the statistics (X2 and OR) only with two decimal places.

Reply:We checked and corrected the title and heading of Table3.

3.Also, a typographical error check is again recommended, as in "sofeware" (Line 148)

Reply: The errors noted above have been corrected in the manuscript.

Reply to reviewer2 

We thank you very much for the constructive comments and suggestions on our manuscript. We have corrected our manuscript according to your comments.

1.Abstract

• Line 43-48- the observed differences between groups (with and without depression need clarifying).

Reply: We checked and corrected the line 43-48 as following “ There were statistical differences between with depression students and no depression students on the rate of the correct answer in the following questions:” in the abstract part of the manuscript.

• Lines 45-53- it is not clear what the dependent variable being predicted here is

Reply:The dependent variable being predicted in Lines 45-49 was “on the rate of the correct answer”.

The dependent variable being predicted in Lines 51-53 was “the coping way”.

• The conclusion is not clear, and none of the statements made here are accessible. For example, does the reference to ‘broadcasting mental health’ infer the importance of sharing whether one has mental health problems? It is not clear.

Reply:We thank you very much for the constructive comments and suggestions on our manuscript. We have corrected conclusion as following “It is necessary to focus on mental health of medical students, especially in female, residence in rural area, poor family relationships, and lower cognitive level of depression knowledge. Medical students were insufficient on depression knowledge and coping styles, efforts that train students know risk of impaired mental health could also improve diagnosis and treatment.”

2.Introduction

• Typos- line 70 page 3- should be ‘depression’ not depressing; line 86 page 4 ‘made no treated history’- should this be ‘had no history of treatment’?

Reply: We agree with this suggestion and modified the two sections in our manuscript.

• The introduction is very brief, and lacking in analysis of the presented information. For example, page 3 line 78 could comment on and explore reasons for the significant differences in prevalence rates (6-66.5%).

Reply: We added some comments in the part of introduction in our manuscript.

3.Method

• Page 4 line 104- should read ‘the questionnaire survey’ or simply ‘the survey’

Reply: We agree with this suggestion. We corrected the express with ‘the questionnaire survey’.

• Arrangements for informed consent are slightly concerning. It is not clear how consent was informed based only on a ‘statement’ particularly given the sensitive nature of the study. Did participants explicitly consent (e.g. by ticking a box) or was consent inferred by progressing with questionnaire? How much information about the study did they receive prior to completing the questionnaires?

Reply: We thank you very much for the constructive comments and suggestions on our manuscript. Arrangements for informed consent as following: “The questionnaire included a statement about informed consent at the beginning. The participant was asked to answer the question “are you still willing to participate in our survey after read and understand our research intentions?” Only those who chose "yes" were considered consent to write out the questionnaire.” 

• Page 6 line 148 typo- should be ‘software’

Reply: The errors noted above have been corrected, and showed in the manuscript.

4.Results

• Table 3 needs clarifying. For example, the % of correct answers column- percentages are in brackets but not clear what the number appearing before this is? Is this number of correct answers? Does the ‘with depression’ and ‘no depression’ colums show the number in each group endorsing a correct answer? This is not clear. Finally was the chi square comparing absolute numbers in each group (depression v no depression) endorsing the correct answer or the proportions? This needs explaining in the main body. It also needs to be clear what the significant results show.

Reply: We checked and corrected the title with “ Number of correct answer about the knowledge of depression in medical students”, which indicated “with depression’ and ‘no depression’ colums show the number in each group endorsing a correct answer” , accordingly, the heading of table3 also been corrected.

• Logistic regression results- the concept of ‘no to coping with’ which is the outcome variable needs clarifying as it is not accessible- i.e. not clear what it means. 

Reply: Only the medical students with depression conducted the Logistic regression analysis, coping styles for depression was the dependent variable. 

• Multiple comparisons increase risk of family-wise error rate. Thus Bonferroni correction is needed for all analyses.

Reply:We agree with this suggestion. Bonferroni correction was used to all analysis in our manuscript.

5.Discussion

• Page 13 line 224-225 is directly contradicting lines 220-222.

Reply: :We agree with your suggestion. The content in lines 220-222 was deleted.

• Lines 225-227- not clear here whether the comparisons being discussed are of other medical students.

Reply: In Lines 225-227, the comparisons being discussed are of other medical students in related many studies. More details can find in our manuscript.

• Line 232- do you mean ‘wellness’

Reply: The errors noted above have been corrected in the manuscript.

• Linbe 268- do you mean ‘will be qualified’?

Reply: The errors noted above have been corrected with “will be able to”.

• Line 270- do you mean ‘could affect’

Reply: The errors noted above have been corrected with “could affect”.

• Line 271-272 (and throughout the paper)- you refer to ‘no coping way’- do you mean ‘no coping mechanisms’? or ‘no ways of coping’?

Reply:We agree with your suggestion. We used the expression “ no ways of coping” in full manuscript.

 • Iine 272-273- it is not clear how your results compare to Thompson as there is a lack of clarity in regards to your conceptualisation of ‘no coping way’.

Reply: We agree with your suggestion.We used the expression “ no ways of coping”, meaning that no any ways to coping with depression or ignore in our manuscript. Our results are similar to Thompson`s.

• Lines 273-281- need revising for grammar

Reply:We agree with your suggestion. We revised the part carefully. More details can be find in our manuscript.

• Lines 292-296- not substantive. Multiple tools can still capture depression adequately, and there is never an expectation for a study to use all available tools

Reply:We agree with your suggestion. We deleted the content in the part of limitation.

• Lines 296-298- need revising for meaning

Reply: The content in lines 296-298 “among the variety of tools available to assess depression, this study used only SDS as a tool to assess symptoms of depression. The SDS scale is considered a good discriminat tool between depressed and non-depressed individuals, only can to compare with studies used the scale, can not compare with studies used other tools.” was deleted according to the full text . 

Once again, thank you very much for your constructive comments and suggestions which would help us both in English and in depth to improve the quality of the paper.

---

## [Decision Letter · Decision Letter 2]

7 Feb 2023

PONE-D-22-10652R2Knowledge and Coping Style About Depression in Medical Students : A Cross-Sectional Study in North ChinaPLOS ONE

Dear Dr. Ping,

Thank you for submitting your manuscript to PLOS ONE. After careful consideration, we feel that it has merit but does not fully meet PLOS ONE’s publication criteria as it currently stands. Therefore, we invite you to submit a revised version of the manuscript that addresses the points raised during the review process.

We look forward to receiving your revised manuscript.

Kind regards,

Nabeel Al-Yateem, PhD

Academic Editor

PLOS ONE

Journal Requirements:

Reviewers' comments:

Reviewer's Responses to Questions

**Comments to the Author**

1. If the authors have adequately addressed your comments raised in a previous round of review and you feel that this manuscript is now acceptable for publication, you may indicate that here to bypass the “Comments to the Author” section, enter your conflict of interest statement in the “Confidential to Editor” section, and submit your "Accept" recommendation.

Reviewer #1: All comments have been addressed

Reviewer #3: (No Response)

2. Is the manuscript technically sound, and do the data support the conclusions?

Reviewer #1: Yes

Reviewer #3: Partly

3. Has the statistical analysis been performed appropriately and rigorously? 

Reviewer #1: Yes

Reviewer #3: Yes

4. Have the authors made all data underlying the findings in their manuscript fully available?

Reviewer #1: Yes

Reviewer #3: Yes

5. Is the manuscript presented in an intelligible fashion and written in standard English?

Reviewer #1: Yes

Reviewer #3: No

6. Review Comments to the Author

Reviewer #1: All comments have been addressed.

I note some style errors that will surely be pointed out when editing the manuscript.

Reviewer #3: I would suggest that the author hires someone to correct the English language. It turns out that the previous version was much better than the update one when it comes to grammar and language editing.

On the titles of the summary tables, it is not clear what the Odds Ratio (OR) is about. Specifically, what is the outcome variable, and what is the model predicting?

I suggest clearly distinguishing between crude OR and adjusted OR.

The selected sample may not represent the entire student population. Thus, the result may not be generalizable.

It would be nice to show the interval reliability measures of the responses.

Some formatting is needed for representing the OR with CI and similar places throughout the paper.

For example, OR with confidence intervals are written as 1.027(0.744,1.418)

There should be a space after 1.027 and after the comma, to look like the following: 1.027 (0.744, 1.418)

7. PLOS authors have the option to publish the peer review history of their article (what does this mean?). If published, this will include your full peer review and any attached files.

Reviewer #1: No

Reviewer #3: No

---

## [Author Response · Author response to Decision Letter 2]

26 Feb 2023

Our responds to the editor’s and the reviewer’s comments：

Reply to editor

We thank you very much for the constructive comments and suggestions on our manuscript. We have corrected our manuscript according to the comments.

Please review your reference list to ensure that it is complete and correct. If you have cited papers that have been retracted, please include the rationale for doing so in the manuscript text, or remove these references and replace them with relevant current references. Any changes to the reference list should be mentioned in the rebuttal letter that accompanies your revised manuscript. If you need to cite a retracted article, indicate the articles retracted status in the References list and also include a citation and full reference for the retraction notice.

 Reply: We review carefully our reference list, it is complete and correct, and no retracted reference were cited.

Reply to reviewer1

Reviewer #1: All comments have been addressed. I note some style errors that will surely be pointed out when editing the manuscript. 

Reply: We thank you very much for the comments and suggestions on our manuscript. We corrected as far as possible in style errors and spelling mistake in our manuscript.

Reply to reviewer2

1.I would suggest that the author hires someone to correct the English language. It turns out that the previous version was much better than the update one when it comes to grammar and language editing. 

Reply: We thank you very much for the comments and suggestions on our manuscript. We invide the English specialized staff to correct our manuscript .

2.On the titles of the summary tables, it is not clear what the Odds Ratio (OR) is about. Specifically, what is the outcome variable, and what is the model predicting? I suggest clearly distinguishing between crude OR and adjusted OR. 

Reply: We thank you very much for the comments and suggestions on our manuscript. We added the note at the bottom of the table as follows “ it means that the size of risk on a characteristic between the medical students with depression and no depression.”

3.The selected sample may not represent the entire student population. Thus, the result may not be generalizable. 

Reply: We thank you very much for the comments and suggestions on our manuscript. We highlighted the piont in the part of limitation as follows “ The medical students in our study were not be selected randomly, so, they didn't represent the entire student population in China, the result may have effect on generalization ”.

4.It would be nice to show the interval reliability measures of the responses. 

Reply: We thank you very much for the comments and suggestions on our manuscript. We calculated the Cronbach's α for interval reliability in our manuscript.

5.Some formatting is needed for representing the OR with CI and similar places throughout the paper. For example, OR with confidence intervals are written as 1.027(0.744,1.418) There should be a space after 1.027 and after the comma, to look like the following: 1.027 (0.744, 1.418)

Reply: We thank you very much for the comments and suggestions on our manuscript. We corrected as far as possible in style errors and spelling mistake in our manuscript.

Once again, thank you very much for your constructive comments and suggestions which would help us both in English and in depth to improve the quality of the paper.

---

## [Decision Letter · Decision Letter 3]

2 May 2023

PONE-D-22-10652R3Knowledge and Coping Style About Depression in Medical Students : A Cross-Sectional Study in North ChinaPLOS ONE

Dear Dr. Ping,

Thank you for submitting your manuscript to PLOS ONE. After careful consideration, we feel that it has merit but does not fully meet PLOS ONE’s publication criteria as it currently stands. Therefore, we invite you to submit a revised version of the manuscript that addresses the points raised during the review process.

We look forward to receiving your revised manuscript.

Kind regards,

Nabeel Al-Yateem, PhD

Academic Editor

PLOS ONE

Reviewers' comments:

Reviewer's Responses to Questions

**Comments to the Author**

1. If the authors have adequately addressed your comments raised in a previous round of review and you feel that this manuscript is now acceptable for publication, you may indicate that here to bypass the “Comments to the Author” section, enter your conflict of interest statement in the “Confidential to Editor” section, and submit your "Accept" recommendation.

Reviewer #3: All comments have been addressed

Reviewer #4: (No Response)

2. Is the manuscript technically sound, and do the data support the conclusions?

Reviewer #3: Yes

Reviewer #4: Partly

3. Has the statistical analysis been performed appropriately and rigorously? 

Reviewer #3: Yes

Reviewer #4: N/A

4. Have the authors made all data underlying the findings in their manuscript fully available?

Reviewer #3: (No Response)

Reviewer #4: Yes

5. Is the manuscript presented in an intelligible fashion and written in standard English?

Reviewer #3: Yes

Reviewer #4: No

6. Review Comments to the Author

Reviewer #3: (No Response)

Reviewer #4: 1.The language used lacks academic rigor and coherence, which affects the clarity and precision of the text.

2.There are errors in the expression of some statistical symbols used in the study, such as the chi-square value. It is recommended to carefully review and correct these errors.

3.The selection of keywords and the phrasing of the title may benefit from further optimization to accurately and effectively communicate the study's focus and contribution.

4.The current research lacks substantial depth in both its content and analysis.

7. PLOS authors have the option to publish the peer review history of their article (what does this mean?). If published, this will include your full peer review and any attached files.

Reviewer #3: **Yes: **Enayetur Raheem

Reviewer #4: No

---

## [Author Response · Author response to Decision Letter 3]

31 May 2023

Our responds to the reviewer’s comments：

Reply to reviewer4

We thank you very much for the constructive comments and suggestions on our manuscript. We have corrected our manuscript according to the comments.

1.The language used lacks academic rigor and coherence, which affects the clarity and precision of the text. 

Reply:We thank you very much for the comments and suggestions on our manuscript. We invide the English specialized staff to correct our manuscript .

2.There are errors in the expression of some statistical symbols used in the study, such as the chi-square value. It is recommended to carefully review and correct these errors.

Reply: We thank you very much for the comments and suggestions on our manuscript. We corrected as far as possible in some statistical symbols used in our manuscript.

3.The selection of keywords and the phrasing of the title may benefit from further optimization to accurately and effectively communicate the study's focus and contribution. 

Reply: We thank you very much for the comments and suggestions on our manuscript. We corrected some keywords in our manuscript.

4.The current research lacks substantial depth in both its content and analysis.

Reply: We thank you very much for the comments and suggestions on our manuscript.We added content in the part of discussion.

Once again, thank you very much for your constructive comments and suggestions which would help us both in English and in depth to improve the quality of the paper.

---

## [Decision Letter · Decision Letter 4]

31 Jul 2023

PONE-D-22-10652R4Knowledge and Coping Style About Depression in Medical Students : A Cross-Sectional Study in North ChinaPLOS ONE

Dear Dr. Ping,

Thank you for submitting your manuscript to PLOS ONE. After careful consideration, we feel that it has merit but does not fully meet PLOS ONE’s publication criteria as it currently stands. Therefore, we invite you to submit a revised version of the manuscript that addresses the points raised during the review process.

We look forward to receiving your revised manuscript.

Kind regards,

Nabeel Al-Yateem, PhD

Academic Editor

PLOS ONE

Journal Requirements:

Reviewers' comments:

Reviewer's Responses to Questions

**Comments to the Author**

1. If the authors have adequately addressed your comments raised in a previous round of review and you feel that this manuscript is now acceptable for publication, you may indicate that here to bypass the “Comments to the Author” section, enter your conflict of interest statement in the “Confidential to Editor” section, and submit your "Accept" recommendation.

Reviewer #5: All comments have been addressed

2. Is the manuscript technically sound, and do the data support the conclusions?

Reviewer #5: Yes

3. Has the statistical analysis been performed appropriately and rigorously? 

Reviewer #5: I Don't Know

4. Have the authors made all data underlying the findings in their manuscript fully available?

Reviewer #5: Yes

5. Is the manuscript presented in an intelligible fashion and written in standard English?

Reviewer #5: Yes

6. Review Comments to the Author

Reviewer #5: Please clarify the number of respondents excluded and the reasons for exclusion to provide transparency in sample selection.

Provide more details about the demographic characteristics (e.g., age range, gender distribution) to enhance the understanding of the study population.

Include mean and standard deviation values for continuous variables (e.g., age, SDS score) in addition to frequencies.

Provide a more comprehensive discussion of the findings, including comparisons with previous studies and an analysis of the implications for medical education and student support services.

Discuss the potential reasons behind the high prevalence of depression among medical students in China, considering factors such as academic pressure, long study duration, and clinical practice stress.

Address the limitations of the study more explicitly, including the potential biases associated with self-reporting surveys and the need for further research in different regions of China.

7. PLOS authors have the option to publish the peer review history of their article (what does this mean?). If published, this will include your full peer review and any attached files.

Reviewer #5: No

---

## [Author Response · Author response to Decision Letter 4]

5 Aug 2023

Reply to reviewer5

We thank you very much for the constructive comments and suggestions on our manuscript. We have corrected our manuscript according to your comments.

1.Please clarify the number of respondents excluded and the reasons for exclusion to provide transparency in sample selection.

Reply:We thank you very much for the comments and suggestions on our manuscript. We added the description about the number of respondents excluded and the reasons in 119-123 line as following: During the period of work, 2200 medical students were invited. Each invited medical students could complete the questionnaire only once. By March 20, a total of 1931 questionnaires had been returned by respondents, and the response rate was 87.8%. 269 invited no returned the questionnaire, the reasons may be disagree or miss the time. Questionnaires that had completed less than 100 seconds were considered unusable, 1909 questionnaires were usable, and 22 were removed from the analysis.

2.Provide more details about the demographic characteristics (e.g., age range, gender distribution) to enhance the understanding of the study population.

Reply:We thank you very much for the comments and suggestions on our manuscript. “age range, gender distribution” have been showed in 176-177 line in our manuscript.

3.Include mean and standard deviation values for continuous variables (e.g., age, SDS score) in addition to frequencies.

Reply:We thank you very much for the comments and suggestions on our manuscript. “mean and standard deviation of age ” have been showed in 176 line. “mean and standard deviation of SDS score” have been showed in table 2 in our manuscript. 

4.Provide a more comprehensive discussion of the findings, including comparisons with previous studies and an analysis of the implications for medical education and student support services. Discuss the potential reasons behind the high prevalence of depression among medical students in China, considering factors such as academic pressure, long study duration, and clinical practice stress.

Reply:We thank you very much for the comments and suggestions on our manuscript. A more comprehensive discussion including you mentioned above have been showed in 249-256 line and 312-314 line in our manuscript.

5.Address the limitations of the study more explicitly, including the potential biases associated with self-reporting surveys and the need for further research in different regions of China.

Reply:We thank you very much for the comments and suggestions on our manuscript. A more comprehensive discussion including you mentioned above have been showed in 318-324 in our manuscript.

Once again, thank you very much for your constructive comments and suggestions which would help us both in English and in depth to improve the quality of the paper.

---

## [Editor Report · Decision Letter 5]

16 Oct 2023

Knowledge and Coping Style about Depression in Medical Students: A Cross-Sectional Study in China

PONE-D-22-10652R5

Dear Dr. Ping,

We’re pleased to inform you that your manuscript has been judged scientifically suitable for publication and will be formally accepted for publication once it meets all outstanding technical requirements.

Kind regards,

Nabeel Al-Yateem, PhD

Academic Editor

PLOS ONE
---

## [Editor Report · Acceptance letter]

18 Oct 2023

PONE-D-22-10652R5 

Knowledge and Coping Style about Depression in Medical Students: A Cross-Sectional Study in China 

Dear Dr. Ping:

I'm pleased to inform you that your manuscript has been deemed suitable for publication in PLOS ONE. Congratulations! Your manuscript is now with our production department. 

Kind regards, 

on behalf of

Dr. Nabeel Al-Yateem 

Academic Editor

PLOS ONE